# A New Perspective on SPME and SPME Arrow: Formaldehyde Determination by On-Sample Derivatization Coupled with Multiple and Cooling-Assisted Extractions

**DOI:** 10.3390/molecules28145441

**Published:** 2023-07-16

**Authors:** Stefano Dugheri, Giovanni Cappelli, Niccolò Fanfani, Jacopo Ceccarelli, Giorgio Marrubini, Donato Squillaci, Veronica Traversini, Riccardo Gori, Nicola Mucci, Giulio Arcangeli

**Affiliations:** 1Industrial Hygiene and Toxicology Laboratory, University Hospital Careggi, 50134 Florence, Italy; 2Department of Experimental and Clinical Medicine, University of Florence, 50121 Florence, Italydonato.squillaci@unifi.it (D.S.);; 3Department of Experimental and Clinical Biomedical Sciences “Mario Serio”, University of Florence, 50121 Florence, Italy; 4Department of Drug Sciences, University of Pavia, 27100 Pavia, Italy; giorgio.marrubini@unipv.it; 5Department of Civil and Environmental Engineering, University of Florence, 50121 Florence, Italy

**Keywords:** solid-phase microextraction, environmental analysis, food analysis, formaldehyde, on-sample derivatization, SPME, SPME Arrow, headspace

## Abstract

Formaldehyde (FA) is a toxic compound and a human carcinogen. Regulating FA-releasing substances in commercial goods is a growing and interesting topic: worldwide production sectors, like food industries, textiles, wood manufacture, and cosmetics, are involved. Thus, there is a need for sensitive, economical, and specific FA monitoring tools. Solid-phase microextraction (SPME), with *O*-(2,3,4,5,6-pentafluorobenzyl)-hydroxylamine (PFBHA) on-sample derivatization and gas chromatography, is proposed for FA monitoring of real-life samples. This study reports the use of polydimethylsiloxane (PDMS) as a sorbent phase combined with innovative commercial methods, such as multiple SPME (MSPME) and cooling-assisted SPME, for FA determination. Critical steps, such as extraction and sampling, were evaluated in method development. The derivatization was performed at 60 °C for 30 min, followed by 15 min sampling at 10 °C, in three cycles (SPME Arrow) or six cycles (SPME). The sensitivity was satisfactory for the method’s purposes (LOD-LOQ at 11-36 ng L^−1^, and 8-26 ng L^−1^, for SPME and SPME Arrow, respectively). The method’s linearity ranges from the lower LOQ at trace level (ng L^−1^) to the upper LOQ at 40 mg L^−1^. The precision range was 5.7–10.2% and 4.8–9.6% and the accuracy was 97.4% and 96.3% for SPME and SPME Arrow, respectively. The cooling MSPME set-up applied to real commercial goods provided results of quality comparable to previously published data.

## 1. Introduction

Formaldehyde (FA) is globally synthesized on a large scale by catalytic oxidation of methanol in the vapor phase [1]. FA is commonly used to produce industrial chemicals [2,3,4], and its aqueous solution (known as formalin) has many applications as a sanitizer and preservative [5].

Human FA exposure can result in acute and chronic toxicity. Concerning FA acute toxicity, skin, eyes, and respiratory tract irritation, skin sensitization, and developmental toxicity have been reported [6,7].

Chronic exposure to FA has been recognized to enhance the risk of asthma [8] and miscarriage [9]. Long-term FA exposure has also been related to nasopharyngeal cancer [10] and increased risk of rare neck and head cancers [11].

The complex scenario of FA toxicity, in addition to its mass use, has recently led to the adoption of new regulations in various production sectors. In 2022, the World Trade Organization (WTO) issued a statement from the European Union (EU) on its plan to regulate both FA and FA-releasing substances in wood-based articles and furniture and the interior of vehicles [12,13]. Moreover, in 2021, the European Commission’s Scientific Committee for Consumer Safety (SCCS) proposed lowering the threshold for cosmetics containing FA-releasing substances from 0.05 to 0.001% [14,15]. Vietnam and the European Commission released specific restrictions concerning FA in textiles [16,17,18]. Recently, it has been reported that formalin is extensively used in several tropical countries as an artificial preservative for food [19], contravening the daily limit of oral exposure to FA from the total diet by the European Food Safety Authority (EFSA), which should not exceed 100 mg FA per day [20]. As for food-contact materials, the current US regulation requires that the amount of FA in melamine resins employed on the surface of food-contact products must not exceed 0.5 milligrams per square inch [21]. Since the Federal Hazardous Substances Act (FHSA) in the US listed FA as a “strong sensitizer” substance, the Consumer Product Safety Commission (CPSC) stated that products containing more than 1% of FA could lead to severe hypersensitivity in humans [22]. Concerning the commercialization of children’s products containing FA, the United States’ regulatory agencies set restrictions [23,24,25]. Because of this heterogeneous scenario, evaluating FA content in commercial products is a key factor for compliance with regulations [26]. In this context, green and more efficient approaches to analytical measurements are also aspects to be considered [27]. Also, miniaturization has been implemented in every analytical field, including exposome investigations [28], resulting in cost and time savings throughout the sampling process. The miniaturization of traditional sample preparation devices for liquid chromatography (LC) and gas chromatography (GC) led to the development of new eco-friendly analytical instruments and methods, limiting the impact of chemicals on the environment.

Moreover, applying automated microextraction techniques (METs) [29,30] allows the lowering of time-consuming manual sample preparations. With the introduction of the first commercial device in 1993 by Supelco (Bellefonte, US), solid-phase microextraction (SPME) [31] has proven to be one of the most applied and versatile METs thanks to continuous technological innovations, especially in recent years [29]. In 2015, the Restek Corporation (Bellefonte, US) updated the SPME technology, proposing the SPME Arrow. This tool is a larger-diameter SPME probe with rugged construction, designed to enhance mechanical durability, with a greater phase volume than standard SPME fibers [32,33]. Thus far, polydimethylsiloxane (PDMS) is one of the most widely available coatings employed in the entrapment of volatile analytes via METs; this is mainly due to the ease of control of the extraction process, as well as the absence of competition among analytes during absorption, making it the best choice in the presence of complex matrices [29]. The recent applications involve coupling SPME with on-fiber derivatization [27,34,35,36]. In particular, several reports described the reaction of carbonyl compounds with *o*-(2,3,4,5,6-pentafluorobenzyl) hydroxylamine (PFBHA) and subsequent separation of PFB-oxime derivatives by GC coupled with different detectors [37,38,39,40,41], showing that the derivatives had outstanding chromatographic properties. Other derivatizing reagents are available on the market, such as 2,4-dinitrophenylhydrazine (DNPH) and 2,2,2-trifluoroethylhydrazine (TFEH). However, although DNPH derivatization affords low limits of detection and low chemical noise in the blank samples [38], hydrazones must be extracted or preconcentrated before analysis [42,43,44]. As for TFEH, it could be used to form the TFE–hydrazone [44,45], but the reagent is more expensive than PFBHA. Only a few articles reported the determination of FA using PFBHA on-sample derivatization and SPME sampling in various matrices and the related GC analysis of the PFB-FA-oxime [46,47,48,49,50,51,52]. It is, however, of some interest to investigate the coupling of headspace (HS) or direct injection (DI) SPME with PFB-oxime derivatives to analyze carbonyl functional groups.

We propose the use of a commercially available cooling-assisted SPME [53,54] together with a multiple SPME (MSPME) technique [55,56] for FA monitoring in real-life samples.

Cooling-assisted SPME enhances the fiber efficiency caption while maintaining efficient stripping from the matrix through two different temperature layers.

To our knowledge, combining the cooling-assisted SPME with MSPME has not been reported. Using this technique, we could apply multiple extraction cycles on FA-containing samples, thus enabling the fully automated routine monitoring of FA content of real-life samples. To achieve this goal, we investigated the key points of the on-sample PFBHA derivatization using both SPME and SPME Arrow PDMS fibers.

The developed cooling MSPME method was applied and tested on different matrices to prove its general applicability. Several critical aspects of the technique, such as pH, derivatizing agent excess, and FA blank signal, are discussed.

## 2. Results and Discussion

### 2.1. Method Optimization

Fundamental aspects regarding the optimization of the method are investigated and presented here; in particular, the derivatization temperature, the concentration of the derivatizing reagent, and the effect of salt are considered.

Firstly, we considered the background solution of FA since both reagent water and derivatizing reagent can represent a potential source of contamination [38,57]. Hudson et al. [58] sparged Milli-Q water with ultrapure Argon for 45 min, lowering the FA levels in blanks by 10%. Other procedures reported in the literature, such as UV irradiation and the addition of hydrogen peroxide, showed opposite results in terms of reduction. As a general consideration, it is impossible to define the outcome of FA lowering from the start, as the effect is strictly dependent on the number of dissolved substances that can be transformed and can cause an increase in its concentration [48,58,59]. We observed the FA depletion of roughly 20% (accounting for 8 ± 3 ng) [60] after irradiating with UV light (254 nm, 7.2 W m^−2^) Milli-Q water for 60 min.

Moreover, we considered previous works employing PFBHA-based on-sample derivatization methods (see Table 1).

In this framework, we set up the method using 100 µL of a 20 mg mL^−1^ PFBHA water solution for 1.85 mL of liquid sample and 18 mL of HS.

As for derivatization temperature, Güneş et al. suggested that an increase in the incubation temperature positively affects the reaction efficiency, which was therefore set to 60 °C [61]. The authors also indicated that there was no visible effect on the analysis derived from the salt type; for this reason, NaCl was chosen.

A customized CTC PAL3 System xyz Autosampler (CTC Analytics AG Industrie Strasse 20 CH-4222, Zwingen, Switzerland) installed online to the GC instrument was used to achieve a fully automated procedure, from sample preparation to injection into the GC apparatus. The instrument described improves productivity, minimizing dead times between samples and ultimately reducing the costs of the analytical assay. The full automation of the system requires minimal operator supervision and can process more samples during each analytical session.

A scheme of the procedure, and a picture of the autosampler GC system, are reported in Figure 1.

### 2.2. SPME Absorption by PDMS Coating

The PDMS absorptive coating was chosen for sampling complex matrices since analytes do not compete, unlike using porous phases such as divinylbenzene [62]. The theory of the liquid phase SPME technique was described in previous works. Wardencki et al. demonstrated that increasing the PDMS thickness enhances the analytes’ recovery [47]. Louch, Motlagh, and Pawliszyn reported that the extraction time is inversely proportional to the square of the coating thickness [63]. Moreover, PDMS is the best sorbent for analytes having molecular mass in the 75–300 Da range [64]. Since the weight of PFB-FA-oxime is 225 Da, PDMS was deemed a suitable sorbent phase. We thus selected 100 µm PDMS SPME fibers (among the commercially available 7, 30, and 100 µm, respectively) and 250 µm PDMS SPME Arrow fibers (among the commercially available 100 and 250 µm, respectively).

Considering the liquid fiber coating’s features, the extraction obeys the rules of liquid–liquid partitioning equilibrium. In a three-phase system composed of a liquid polymeric coating, a gaseous headspace, and an aqueous medium, the mass of analyte absorbed (using HS technique) by a coating at equilibrium, *n* (µg), is calculated by Equation (1) [65]:(1)$$n=\frac{C_{o}\times V_{1}\times V_{2}\times K_{1}\times K_{2}}{K_{1}\times K_{2}\times V_{1}+K_{2}V_{3}+V_{2}}$$
where *K*_1_ is the SPME coating/HS partition coefficient, *K*_2_ is the HS/aqueous matrix partition coefficient, *C*_0_ is the initial concentration of the analyte in the aqueous solution (µg mL^−1^), and *V*_1_, *V*_2_ and *V*_3_ are the coating, the aqueous solution, and the HS volumes (mL), respectively.

In the case of DI-SPME sampling from an aqueous medium, Equation (1) can be simplified using Equation (2):(2)$$n=\frac{C_{0}\times V_{1}\times V_{2}\times K}{KV_{1}+V_{2}}$$
where *K*, defined as *K*_1_*·K*_2_, is the partition coefficient between the SPME liquid polymeric coating and the sample.

Pacenti et al. [66] indicated K_ow_ as a good estimator of K. Moreover, K_2_ can be described by means of Equation (3):(3)$$K_{2}=\frac{K_{H}}{R\times T}$$
where *K_H_* is Henry’s constant (mol atm^−1^ L^−1^), *R* is the universal gas constant (L atm K^−1^ mol^−1^), and *T* is the sampling temperature (K, in Kelvin scale).

SPME performs extraction at equilibrium, and therefore it is not exhaustive. Hence, the hypothesis of ideal conditions needed by mathematical modeling must be verified. The distribution constants estimated from physico-chemical tables or by the structural unit contribution method can anticipate trends in SPME analysis; in particular, K_1_ and K_2_ values suggest whether the DI or HS mode is advantageous [52]. To calculate the theoretical mass, using Equations (1) and (2), we considered the same values employed in the method presented, i.e., 20 µg mL^−1^ calibration level (normalized for the final volume and converted in PFB-FA-oxime), V_2_ and V_3_ as 2.5 mL and 17.5 mL, respectively. Physico-chemical constants of the PFB-FA-oxime were obtained by Performs Automated Reasoning in Chemistry (SPARChem, Danielsville, GA, USA). Theoretic recoveries were calculated considering the reaction between immediate and exhaustive FA and PFBHA [37]. The calculation provided *n* values of 200 and 197 µg and theoretical recoveries of 68 and 67% for HS and DI, respectively.

To confirm these theoretical results, we propose the comparison between HS- and DI-SPME using SPME and SPME Arrow fibers. The chromatogram is shown in Figure 2 (the comparison calibration curves are shown in the Appendix A).

As shown in Table 2, the two techniques provide comparable sensitivity (i.e., slope), recoveries, and precision, as found in previous works [49,67]. In more detail, Table 2 shows that the method’s sensitivity was similar using either DI or HS. Exposure time with DI was lower than using HS, while the recoveries were comparable between the two techniques, as observed previously [52].

With water as the medium, DI causes a reduction of the GC column lifetime in comparison with HS. The mean recovery values obtained depend on the non-exhaustive nature of SPME and the inverse proportionality between temperature and captured analyte amount found using each absorbent phase. Equations (1) and (2) show an expectable overestimation of recovered PFB-FA-oxime compared to the experimental values due to the non-exhaustive inherent nature of the process of absorption of the analyte into the sorbent phase. The methods’ precision, expressed by the average CV% of Table 2, was good and found to be similar for the two techniques. Mean recovery is the mean value of the recoveries found for each of the five calibration levels obtained by the PFB-FA-oxime hexane solutions regression curve. The method’s precision was computed by the average variation coefficient (Average CV%) at each calibration level of the curves. Considering these results, we selected HS-SPME as the more suitable technique for determining FA.

### 2.3. Cooling-Assisted SPME

In HS-SPME, heating the sample solution is beneficial since it increases the HS concentration of volatile analytes. However, the increase in sampling temperature decreases the amount of analytes trapped in the fiber coating because the absorption process is exothermic. Given these premises, cooling-assisted SPME is designed to effectively extract the analytes into the HS while improving SPME sensitivity. This technique consists of heating the sample matrix while cooling the sampling fiber, thus simultaneously increasing the sample matrix/HS and HS/fiber coating distribution constants.

Cooling-assisted SPME is very efficient in complex matrices such as sludge, soil, and clay, where the analytes are firmly adsorbed to the active sites of the sample medium [68,69,70].

The cooling-assisted extraction has been described and classified into three approaches: (i) internally cooled SPME using liquid CO_2_ directly on the fiber; (ii) internally cooled fiber based on a thermoelectric cooler (TEC); (iii) externally cooled SPME using circulating fluids (alcohol or cold water), where the HS of the sample vial is cooled from the outside [55,69,71,72,73,74,75].

In the present study, considering the multivariate nature of the underlying phenomena involved in the cooling-assisted HS-SPME, a rapid study of the operating conditions of the SPME and SPME Arrow devices was carried out by applying a 2^3^ full factorial experimental design. We planned eight experiments, adding three additional tests to validate the models computed (the experimental plan with responses is given in the Appendix A). The responses selected were the peak area of PFB-FA-oxime, and the factors studied were the exposure temperature studied in the range from 10 to 20 °C, the fiber exposure time in cooling mode in the range from 15 to 30 min, and the sampling temperature of the derivatized analyte studied in the range from 60 to 80 °C. The models computed were validated in the experimental domain center point (15 °C, 22.5 min, and 70 °C) at the 95% and 99% confidence levels with satisfactory relative error percentages in prediction (8% for conventional SPME and 17% for SPME Arrow). The models for the two types of fiber showed that, within the ranges evaluated, the only significant effect on the process is produced by the fiber exposure temperature, which must be kept at the lower level investigated (10 °C). The remaining two factors have non-statistically significant and negligible effects compared to the former. Therefore, the sampling temperature and the fiber exposure time can be set at any convenient value within the variation range investigated. We thus selected the following operating conditions: sampling temperature at 60 °C, 15 min time, and 10 °C fiber exposure temperature.

Calibration curves were analyzed and compared to traditional HS sampling described in the previous paragraph using a fully automated system, operating as described in point (iii) of the above classification. Table 3 shows cooling HS performances on calibration levels for both SPME and SPME Arrow: the sensitivity expanded compared to HS (2.6 × 10^5^ and 3.7 × 10^5^, as opposed to 8.7 × 10^4^ and 2.4 × 10^5^, for SPME and SPME Arrow, respectively), as well as the recovery (41.7% and 60.5%, as opposed to 14.8% and 38.3%, for SPME and SPME Arrow, respectively), while the average CV% does not show a significant difference from traditional HS curves (8.9% and 7.6%, compared to previous 9.1% and 8.9%, for SPME and SPME Arrow, respectively) (the calibration curves are showed in the Appendix A).

### 2.4. MSPME Extraction

The time used to reach the partition equilibrium in SPME sampling depends on several parameters, such as sample matrix, sample agitation, temperature, and properties of the coating/analyte [76]. An equilibrium time of 30–60 min is very common for SPME sampling [27,66]; on the contrary, extraction time can be shorter, but this leads to lower extraction yields and, therefore, higher detection limits. Generally, a compromise between extraction time and yield is mandatory; yet, specific adjustments can be introduced to reach a higher throughput or an increased sensitivity.

In this framework, MSPME is a rugged procedure suitable for magnifying the analyte’s response in quantitative analyses of complex matrices [77,78,79,80]; it is based on calculating the mass extracted using the peak areas of a few consecutive extractions from the same sample [53,54,77]. The extraction can also be performed using multiple fibers in a single chamber containing the sample, sequentially desorbed in the GC injection block, trapping the volatile compounds at the beginning of the GC column using a cryoscopic technique, before performing a ‘single shot’ chromatographic run.

A practical calculation of the total area can be performed via Equation (4):(4)$$A_{T}=\frac{A_{1}}{1-\beta }$$
where *A*_1_ is the peak area in the first extraction and *β* is calculated from the linear regression of the logarithms of the individual peak areas, as shown in Equation (5) [80]:(5)$$ln\;A_{i}=\left(i-1\right)\times ln\;ln\;\beta +ln\;ln\;A_{1}\;$$
where *A_i_* is the peak area obtained in the *i*th extraction. To obtain a linear trend in the logarithm of peak areas, Tena et al. suggested that the analyte loaded on the fiber must be considered in relation to its concentration—β has an influence since it should be below 0.95 to achieve at least a 5% difference in two consecutive areas, whereas values below 0.4 allow for calculating the analyte simply with a sum of the areas since four extractions provide a recovery above 97% [54].

The MSPME approach was implemented in the cooling HS and tested on calibration levels to achieve complete extraction of FA. Figure 3 shows the peak area depletion performing consecutive extractions on the calibration levels. Table 4 reports the mean recovery, mean β, the CV% range, and the R^2^ on the five calibration levels (the calibration curves obtained with both SPME and SPME Arrow are shown in the Appendix A). The approach resulted in a mean recovery of 97.4% for SPME and 96.3% for SPME Arrow, confirming the feasibility of the method to reach an exhaustive stripping of FA from the sample and, therefore, high sensitivities. The calculated value of β for SPME Arrow, i.e., 0.41, confirms its higher efficiency in trapping the PFB-FA-oxime in our experimental conditions: in fact, three extractions reduce the concentration of the analyte in the HS to blank levels. The maximum CVs% are 10.2% for SPME and 9.6% for SPME Arrow, corresponding to the sixth and third extraction, respectively. Conversely, the time necessary to perform a complete sample analysis increases from about 100 min to about 410 min for SPME and to 215 min for SPME Arrow.

The results obtained on standard solutions confirm that the combination of HS sampling with cooling MSPME represents the best set-up for sampling PFB-FA-oxime following on-sample derivatization.

The method, optimized using cooling and MSPME approaches, is characterized by enhanced performances compared to the on-sample PFBHA derivatization with the initially presented HS sampling approach, as shown in Table 5. LOD was calculated by multiplying 3.3 by the ratio between the standard deviation of blanks and the intercept of the curve, and the LOQ is three times the LOD. The implementation of these two procedures led to a lowering in the LOD and LOQ values from 22 ng L^−1^ (LOD)–73 ng L^−1^ (LOQ) and 14 ng L^−1^ (LOD)–46 ng L^−1^ (LOQ) to 11 ng L^−1^ (LOD)–36 ng L^−1^ (LOQ) and 8 ng L^−1^ (LOD)–26 ng L^−1^ (LOQ), for SPME and SPME Arrow, respectively. The LOD obtained is limited by the FA content in Milli-Q water, which can not be further lowered. The results obtained are comparable with the previous literature studies for the determination of carbonyl compounds [47,49,58,61] using PFBHA on-sample derivatization; nonetheless, the implementation of the newly proposed automated analytical tool leads to a higher sensitivity, albeit requiring more analysis time due to the multiple extraction steps.

Further modifications can be introduced to obtain high analytical throughput, whether or not it is necessary to process a large batch of samples. Louch et al. suggested that the extraction time, i.e., the diffusion time through the watery layer, is proportional to the square migration extent and inversely proportional to the water diffusion coefficient [63]. In particular, by reducing the vial diameter by a factor of three, the authors achieved a decrease in extraction time of an order of magnitude. Furthermore, the higher amount of phase in SPME Arrow allows to perform fewer extraction cycles and can therefore represent a valid choice to improve throughput. Hence, further tests should be conducted to customize the method depending on the analytical requirements. As far as sensitivity is concerned, Cancilla et al. [81] reported that PFB-FA-oxime formation in water differed only in reproducibility, varying the pH, whereas, for higher-molecular-weight aldehydes, yields increased at pH < 3. This observation is justified by the greater availability of the unprotonated hydroxylamine, involved in the first step as pH increases, while the ease of removal of the protonated carbonyl oxygen, i.e., the second step, is enhanced when pH decreases [58,82]. A modification in the pH value can therefore be considered both when the investigated matrix is not responsive enough to the derivatization and to expand the presented method to higher-molecular-weight aldehydes.

### 2.5. Real Samples Analysis

The method developed was applied to different matrices to prove its ruggedness and feasibility. Samples of green apple, plum, tomato, shampoo, and face wash were prepared as described in the Material and Methods section and analyzed via the cooling MSPME approach (Figure 4). Table 6 reports the FA content for each product, the average CVs% on untreated samples, and the CV% range for the constructed curves, for both SPME and SPME Arrow. FA values were calculated from PFB-FA-oxime data obtained by interpolation of both cooling MSPME and cooling MSPME Arrow curves. Generally, the results found for the matrices examined are in accordance with the previous literature data [19,83,84], confirming the method’s suitability for determining FA content in different commercial goods. Appendix A report the curves constructed in matrices investigated using the standard addition method for SPME and SPME Arrow, respectively; the linearity observed in standards slightly worsened, most likely due to the complexity of matrices.

## 3. Materials and Methods

### 3.1. Chemical and Reagents

*O*-(2,3,4,5,6-pentafluorobenzyl) hydroxylamine hydrochloride (PFBHA·HCl) (CAS 57981-02-9), *n*-hexane (CAS 110-54-3), 1-bromo-4-fluorobenzene (CAS n. 460-00-4) and *p*-fluorobenzaldehyde (CAS n. 459-57-4) were purchased from Sigma-Aldrich (Saint Louis, MO, US). Formaldehyde *O*-(pentafluorobenzyl)oxime (PFB-FA-oxime) (CAS 86356-73-2) was purchased from GiottoBiotech (Sesto Fiorentino, Italy). Milli-Q water 18 MΩ cm (mQ), further purified to eliminate FA using PURE UV3—4-Stage UV Water Purification System (Pure n Natural Systems Inc., Steamwood, IL, USA), was obtained from Millipore (Darmstadt, Germany). Helium (99.999%) as GC carrier gas was obtained from Air Liquid (Paris, France). For automation of the SPME on-fiber PFBHA derivatization, HeadSpace screw-top 20 mL glass vials (HSV) (Part No: 5188-2753) and Hdsp cap 18 mm magnetic PTFE/Sil (Part No. 5188-2759) were purchased from Agilent Technologies (Santa Clara, CA, USA).

We purchased 23-gauge 100 µm PDMS FFA-SPME fibers (9.40 mm^2^ phase area, 600 µL phase volume) and 1.5 mm PDMS 250 µm FFA-SPME Arrow fibers (62.8 mm^2^ phase area, 11.8 µL phase volume) from Chromline (Prato, Italy).

### 3.2. Samples

Fruits samples were purchased in a local market and were green apples (Renette variety), plums (Prunus Domestica Black Amber variety), and tomatoes (Ciliegino and Pachino varieties). Cosmetic products included one shampoo sample and one all-purpose face wash sample.

### 3.3. PFBHA On-Sample Derivatization Routine and Online SPME Sampling

The FA working solution was prepared at 80 mg L^−1^ in water by diluting a 4% (m/m) stock solution.

All steps of the procedure described in the following were fully automated.

On-sample derivatization was performed at 60 °C for 30 min on five calibration levels, dispensing 0, 125, 250, 500, and 1000 µL of the working solution, respectively, into 2 mL vials, containing variable amounts of water, a 20 mg mL^−1^ PFBHA water solution (100 µL) and 50 µL of 20 mg mL^−1^ of *p*-Fluorobenzaldehyde in ethanol.

*p*-Fluorobenzaldehyde was used as IS according to its conformity for SPME-GC analysis of carbonyl compounds derivatized with PFBHA [49,67]. FA final concentrations obtained were 0, 5, 10, 20, and 40 mg L^−1^. The reacted mixtures (2 mL) were then transferred into a 20 mL HSV, containing 1 g of NaCl and 50 µL of 100 mg L^−1^ 1-bromo-4-fluorobenzene in water solution (IS of process, ISP, 50 µL).

1-Bromo-4-fluorobenzene was used as IS in agreement with Güneş et al. [61], given its retention time in proximity to the PFB-FA-oxime, as a quality check for the subsequent analytical steps

Samples were prepared in a 2 mL HSV by diluting each product into proper quantities of water to a final volume of 1.8 mL. Fruit samples peeled and cut into small pieces, were blended in a 1:10 ratio with water. The juice was filtered through a pleated paper filter.

Shampoo and face wash samples were diluted in water (0.8 g in 1 mL of water) and mixed in an ultrasonic bath for 15 min at room temperature, as indicated by Feher et al. [83].

The subsequent on-sample derivatization was performed by adding 20 mg mL^−1^ PFBHA water solution (100 µL) and 20 mg mL^−1^ ISD (50 µL), at 60 °C for 30 min under stirring; the reacted mixtures were transferred in 20 mL HSV containing NaCl and ISP in the proportion indicated above. Four additional vials were prepared in full automation using the same procedure for each commercial product tested, adding 125, 250, 500, and 1000 µL of the FA working solution, respectively, to the initial mixture.

For HS sampling, the fiber was exposed for 30 min to the HS of samples/standards, previous equilibrium under stirring at 60 °C for 20 min, and desorbed in the injector at 250 °C for 2 min.

As for the DI mode, derivatization was performed at 60 °C for 30 min on five calibration levels, prepared following the same proportion used for HS curves, using a “one-pot” approach, in 20 mL HSV. The fiber was dipped for 20 min in the solution, previous equilibrium for 5 min under stirring, and desorbed in the injector at 250 °C for 2 min.

Calibrators and authentic samples were prepared independently and analyzed five-fold in random order.

Regarding the cooling MSPME approach, the same sets of reference standards and samples were prepared following the HS procedure. Consecutive repetitions were performed on each HSV operating in MSPME mode for both standards and samples. The parameters were fiber exposure temperature set at 10 °C, reacted mixture temperature set at 60 °C, 15 min time of exposure, 6 extraction cycles for SPME, and 3 extraction cycles for SPME Arrow.

### 3.4. Online Robotic System

Automation of the analytical procedure was achieved using a CTC PAL3 System xyz Autosampler (CTC Analytics AG Industrie Strasse 20 CH-4222, Zwingen, Switzerland) with a 1200 mm bar. The apparatus was equipped with a Multi Fiber eXchange (MFX) system (Chromline, Prato, Italy), Liquid Syringe Tool (CTC Analytics AG, Zwingen, Switzerland), SPME dual layer extraction (SDLE) equipped with Chronos software (Chromline, Prato, Italy), an FFA-SPME holder, a tray with 20 mL and 2 mL slots, a fiber conditioning system, two solvent modules (one for 100 mL solvent bottles and the other for 10 mL solvent bottles), and a wash module, to guarantee an automated routine between the exchange of syringe, FFA-SPME, and FFA-SPME Arrow fibers, and shift between traditional SPME and cooling SPME sampling.

### 3.5. GC-MS Operating Conditions

The chromatographic method proposed by Dugheri et al. was used [60], with a Varian CP3800 GC system coupled to a Varian Saturn 2200 Ion-Trap as the detector (scan mode, 45–300 m/z, EI energy 70 eV).

The column was a DB 35-MS-UI (Agilent J&W), and the 1079 injector port (SCION, Instruments, Amundsenweg, The Netherlands) was provided with a 0.75 mm internal diameter liner. The oven settings were isotherm of 50 °C for 1 min, followed by a linear temperature ramp of 10 °C min^−1^ to 260 °C. Helium was used as the carrier gas, set at a flow rate of 1.2 mL min^−1^. The absolute quantity of PFB-FA-oxime was calculated on a regression curve obtained via automatic direct injection (1 μL) of hexane solutions in the GC system (0, 5, 10, 20, and 40 mg L^−1^ of PFB-FA-oxime, respectively) to assess the recovery of the method. When operating in cooling SPME mode, the SDLE module, installed on the PAL 6-position agitator, was set at 10 °C using Chronos software.

## 4. Conclusions

The quantitative determination of FA in an aqueous medium is possible by PFBHA on-sample derivatization and HS- or DI-SPME extraction using a PDMS fiber prior to GC separation and MS detection. The implementation of cooling and multiple SPME approaches to the method leads to satisfactory performances of the methods in terms of sensitivity and recovery. The full automation of sample preparation and analysis makes the procedure well-suited for the routine determination of FA.

Optimal extraction conditions were selected using a 20 mg mL^−1^ PFBHA water solution, followed by 30 min of derivatization time and subsequent SPME or SPME Arrow extraction, with a PDMS coating, for 15 min. Under these conditions, the LOD and LOQ methods were 11 and 36 ng L^−1^ and 8 and 26 ng L^−1^ for SPME and SPME Arrow, respectively. The methods’ response functions showed good linearity over the ranges tested, i.e., LOQ of 40 mg L^−1^, CV% ranging from 5.7 to 10.2 and 4.8 to 9.6, and FA recoveries of 97.4% and 96.3% for SPME and SPME Arrow, respectively.

The method’s effectiveness was also demonstrated by testing fruits and cosmetic products. The results obtained confirmed the general suitability of the method proposed here for routine analysis of real-life samples.

## Figures and Tables

**Figure 1 molecules-28-05441-f001:**
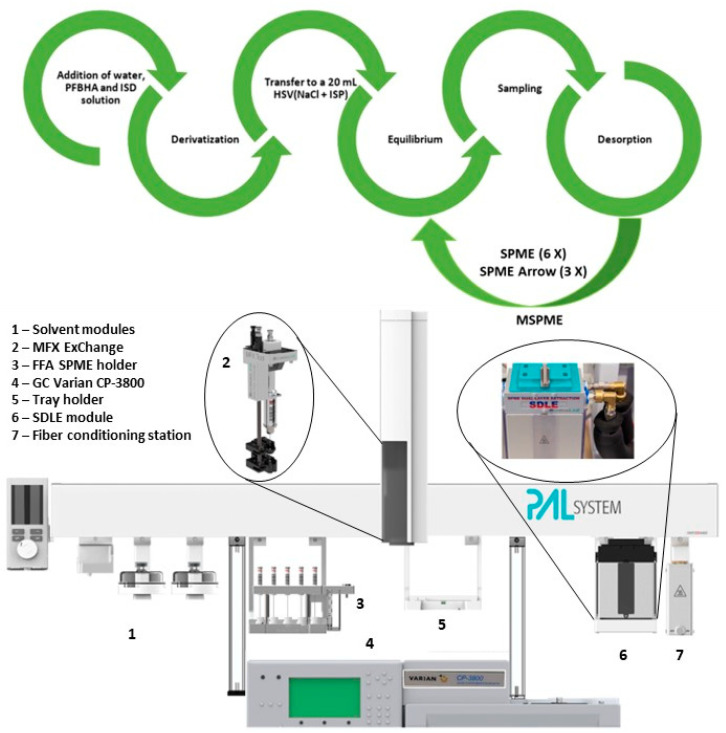
Flow chart and image of autosampler for the fully automated on-sample derivatization and analysis of PFBHA FA derivative with internal standardization (Picture courtesy of Filippo Degli Esposti, Chromline, Prato, Italy, free domain).

**Figure 2 molecules-28-05441-f002:**
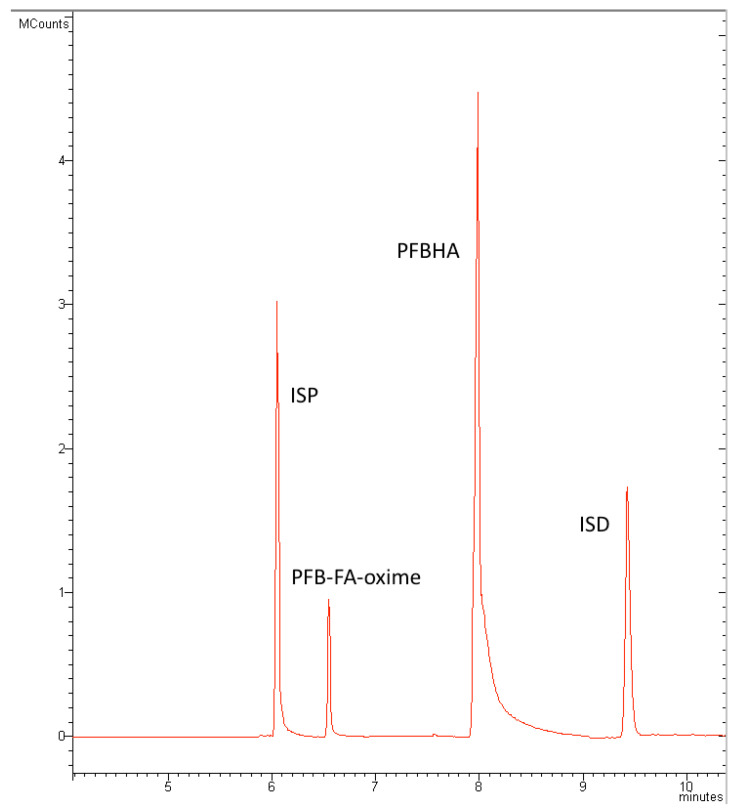
Total ion current chromatogram in EI mode, showing ISP, PFB-FA-oxime, PFBHA, and ISD.

**Figure 3 molecules-28-05441-f003:**
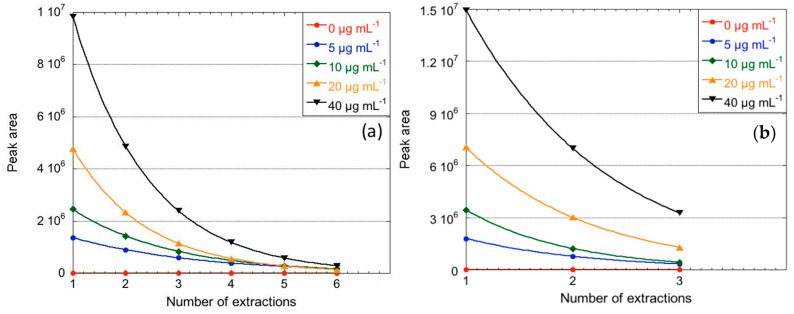
Decrease in peak area with the number of injections on calibration levels for SPME (**a**) and SPME Arrow (**b**).

**Figure 4 molecules-28-05441-f004:**
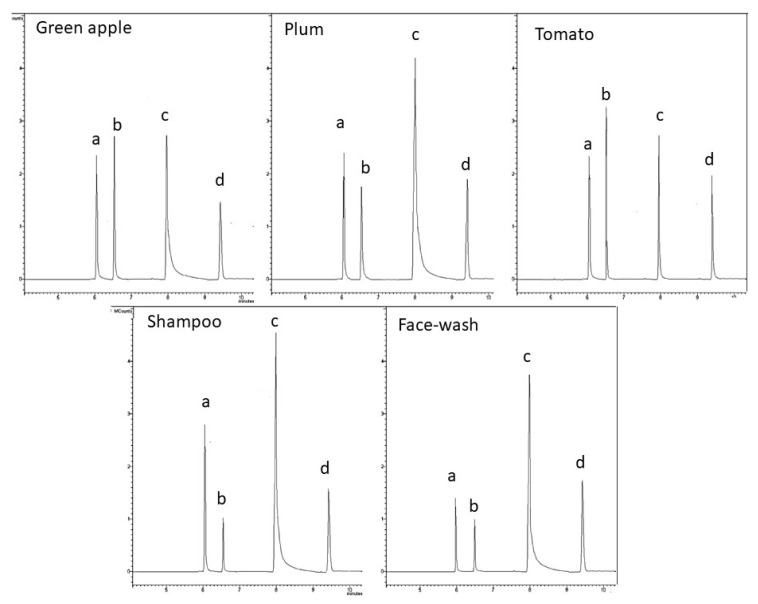
Chromatograms of real samples (green apple, plum, tomato, shampoo, and face wash) analysis: ISs (a and d), PFB-FA-oxime (b), and PFBHA (c) peaks.

**Table 1 molecules-28-05441-t001:** The literature conditions for PFBHA-based on-sample derivatization methods.

PFBHA Solution Concentration (mg/mL)	PFBHA Solution Volume Used (µL)	Derivatization Target	Sample Volume (mL)	Sample Matrix	Reference
12	50	Acetone	10	Seawater	[58]
12	100	Aliphatic aldehydes C1 to C9 (from formaldehyde to nonanal), acetone, acrolein, butanone, furfural, benzaldehyde, methylglyoxal, glyoxal, 2,4-pentane dione	20	Seawater	[48]
2	1000	Formaldehyde, acetaldehyde, acetone, propionaldehyde, acrolein, isobutyraldehyde, butyraldehyde, pentanal, crotonaldehyde, isovaleraldehyde, hexanal	9	Alcoholic beverages	[47]
6	40	23 carbonyl compounds, including C1–C10 saturated aliphatic and unsaturated aldehydes, ketones, and dialdehydes	4.0–8.5	Water	[49]

**Table 2 molecules-28-05441-t002:** Comparison of the slope ± standard error, exposure time, mean recovery, average CV%, and R^2^ between HS and DI using SPME and SPME Arrow.

	SPME		SPME Arrow	
	HS	DI	HS	DI
Slope ± Standard Error	(8.7 ± 0.1) × 10^4^	(7.6 ± 0.1) × 10^4^	(2.37 ± 0.04) × 10^5^	(2.01 ± 0.08) × 10^5^
Intercept ± Standard Error	(1.0 ± 2.5) × 10^4^	(3.4 ± 3.0) × 10^4^	−(1.0 ± 0.9) × 10^4^	(1.7 ± 1.5) × 10^5^
Exposure time (min)	30	20	30	20
Mean recovery (%)	14.8	14.2	38.3	37.4
R^2^	0.9994	0.9989	0.9990	0.9962
Average CV%	9.1	10.3	8.9	9.7

**Table 3 molecules-28-05441-t003:** Slope and intercept ± standard error, exposure temperature, mean recovery, average CV%, and R^2^ for cooling HS using SPME and SPME Arrow.

	Cooling SPME	Cooling SPME Arrow
Slope ± Standard Error	(2.44 ± 0.03) × 10^5^	(3.74 ± 0.07) × 10^5^
Intercept ± Standard Error	(2.7 ± 7.1) × 10^4^	−(1.5 ± 1.4) × 10^5^
Exposure temperature (°C)	10	10
Mean recovery (%)	41.7	60.5
R^2^	0.9994	0.9991
Average CV%	8.9	7.6

**Table 4 molecules-28-05441-t004:** Slope and intercept ± standard error, mean recovery, mean β values, CV% range, and R^2^ for the calibration levels, studied with cooling MSPME and cooling MSPME Arrow techniques.

	Cooling MSPME	Cooling MSPME Arrow
Slope ± Standard Error	(4.8 ± 0.1) × 10^5^	(6.32 ± 0.05) × 10^5^
Intercept ± Standard Error	(0.1 ± 2.0) × 10^5^	−(2.0 ± 1.0) × 10^5^
Mean recovery (%)	97.4	96.3
β	0.56	0.41
CV% range	5.7–10.2	4.8–9.6
R^2^	0.9990	0.9998

**Table 5 molecules-28-05441-t005:** Comparison between the performances obtained in this work and previous studies.

	Column	Technique	LOD	LOQ
Cooling MSPME	Fused-silica 35% Ph (30 m × 0.25 mm, 0.25 μm film thickness)	GC-MS	11 ng L^−1^	36 ng L^−1^
Cooling MSPME Arrow	Fused-silica 35% Ph (30 m × 0.25 mm, 0.25 μm film thickness)	GC-MS	8 ng L^−1^	26 ng L^−1^
Bao et al. [49]	Fused-silica capillary column (30 m × 0.25 mm I.D., 0.25 μm film thickness)	SPME-GC-ECD	20 ng L^−1^	-
Gunes et al. [61]	Wax capillary column (30 m × 0.32 mm ID, 0.25 μm film thickness)	GC-FID	50 µg L^−1^	167 µg L^−1^
Hudson et al. [58]	5%-phenyl-methylpolysi-loxane (30 m × 0.25 mm id × 0.25 μm film thickness)	SPME-GC-MS	3.0 ng L^−1^	-
Wardencki et al. [47]	Fused-silica containing Rtx-5 (30 m × 0.32 i.d., 3 μm film thickness)	SPME-GC-ECD	5 ng L^−1^	-

**Table 6 molecules-28-05441-t006:** Mean FA content, R^2^, average CV% and CV% range for each matrix investigated.

Matrix	Mean FA Content (mg/L)		R^2^		Average CV%		CV% Range	
	SPME	SPME Arrow	SPME	SPME Arrow	SPME	SPME Arrow	SPME	SPME Arrow
Green apple	11.8	11.2	0.9873	0.9990	7.5	6.6	6.8–9.8	5.3–8.8
Plum	9.16	11.4	0.9939	0.9969	6.2	5.9	5.3–8.5	5.2–9.2
Tomato	14.5	20.8	0.9841	0.9930	8.1	5.3	6.2–9.6	4.2–7.3
Shampoo	3.51	6.9	0.9984	0.9882	7.0	4.2	5.9–8.2	3.9–6.7
Face-wash	4.92	8.0	0.9901	0.9987	7.6	4.5	6.5–8.4	3.8–6.9

## Data Availability

Not applicable.

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
