# Peer review of "A New Perspective on SPME and SPME Arrow: Formaldehyde Determination by On-Sample Derivatization Coupled with Multiple and Cooling-Assisted Extractions"

_molecules, 2023, doi:10.3390/molecules28145441_

Round 1

Reviewer 1 Report (Previous Reviewer 2)

 1: In the TOC figure, the words for the chromatographic peak can not be seen clearly.

2:  Table 1 is not normative and rigorous. It should be redesigned with a three-line table.

3:  The analytical concentration (ug/mL) in Figure 2 is very high. The authors should pay more attention to trace analysis. Furthermore, such a figure of standard curve might not be listed in the text.

4:  Some Figures and Tables should be placed in the Supporting Information. The chromatography for trace FA analysis should be presented in the text, especially corresponding chromatograms for Table 6.

5:  Figure 6 is very important for this paper, but why it was set as the last figure? In Figure 6, there are so many arrows and words on the top.

Author Response

Reviewer 2 Report (Previous Reviewer 4)

The authors report a new sensitive, economical, and specific method for monitoring of formaldehyde in food and cosmetics samples. As the previous time my first impression is excellent. Very well designed and performed an experiment with a wide area of application.

After careful reading of the manuscript, I can conclude that the manuscript looks much better with improved quality.  The additional photos make it much more informative and easy to understand the text. 

I do not have any critical remarks on the manuscript and I would like to conclude and suggest that the manuscript can be published the way it is.

Author Response

Thank you for your comments to our work; we are glad you appreciate our paper.

Round 2

Reviewer 1 Report (Previous Reviewer 2)

The authors have revised the manuscript as the suggestions. It can be accepted.

This manuscript is a resubmission of an earlier submission. The following is a list of the peer review reports and author responses from that submission.

Round 1

Reviewer 1 Report

Introduction may be too long for an original article. It requires to be no more than a page or a page with a half long.

Author Response

“Introduction may be too long for an original article. It requires to be no more than a page or a page with a half long.”

Thank you for your suggestion. We have shortened the introduction, keeping the main informations as suggested.

Reviewer 2 Report

Formaldehyde is a toxic compound. There are so many reports and detection methods about FA and FA-releasing substances analysis. I just think this paper is a regular analysis and testing of samples, but not a novel investigation. Especially, so many standard curves and simple figures, but not important results, are presented. I really think this paper is not suitable for this journal. Another analytical journal may be a better choice. Thanks.

 Author Response

“Formaldehyde is a toxic compound. There are so many reports and detection methods about FA and FA-releasing substances analysis. I just think this paper is a regular analysis and testing of samples, but not a novel investigation. Especially, so many standard curves and simple figures, but not important results, are presented. I really think this paper is not suitable for this journal. Another analytical journal may be a better choice. Thanks.”

Thank you for your comments. Following you suggestion, we have highlighted in the text the novelty of our application: this paper represents the first article where one SPME-related analytical tool, SPME dual layer extraction (SDLE), commercialized in 2022, is integrated in a fully automated procedure, combined with a multiple SPME extraction. These new device was tested and chemometrically optimized to analyze formaldehyde, but its application to absorbent phases as PDMS can be wider. We strongly believe that the study conducted is mandatory to assess the performances of the micro-extraction technique used: our results show very high recoveries and sensitivity, confirmed by the application of our method to real matrices. Moreover, the same experimental steps could be extended to other micro extraction devices available on the market (Hi-sorb, Monotrap, Sorbent Pen, Thin Film, etc.), to overcome competition between analytes.

Reviewer 3 Report

Overall, this is a clear and concise manuscript; the methods are generally appropriate and has implications for the theoretical basis; the results are clear and compelling. Specific comments follow.

(1)The author(s) concluded that high sensitivity was achieved using the proposed approach. The comparison of sensitivity obtained in the literatures should be provided. If the sensitivity of this work can just be comparable to the published paper in J. Chromatogr. A 1998 (ref. 49) and Food Chem. 2019 (ref. 67), the conclusion could be modified.

(2)p.3, line 116-117:The most recent applications…[27,34-36]. The reference 34 is Acta Chromatogr. 2009 and reference 36 is Bull. Korean Chem. Soc. 2013, which could be replaced by more suitable and recent literatures.

(3)Please check and correct the order of concentrations in the note of Figure 4.(b).

Author Response

“(1)The author(s) concluded that high sensitivity was achieved using the proposed approach. The comparison of sensitivity obtained in the literatures should be provided. If the sensitivity of this work can just be comparable to the published paper in J. Chromatogr. A 1998 (ref. 49) and Food Chem. 2019 (ref. 67), the conclusion could be modified."

Thank you for your comment. SPME and SPME Arrow techniques provided good accordance with the two references indicated, in terms of recovery and precision, in HS mode. As for the conclusions, the “high sensitivity” we indicated is referred to the improvement provided by cooling and Multiple extraction techniques, which enhanced the slopes initially obtained by using conventional HS extraction. We explained this in the text as well: “The implementation of cooling and multiple SPME approaches to the method leads to an enhancement of the performances, in terms of sensitivity and recovery”.

"(2)p.3, line 116-117:The most recent applications…[27,34-36]. The reference 34 is Acta Chromatogr. 2009 and reference 36 is Bull. Korean Chem. Soc. 2013, which could be replaced by more suitable and recent literatures."

Thank you for your comment. We have replaced the references 34 and 36, with more recent ones.

"(3)Please check and correct the order of concentrations in the note of Figure 4.(b)."

Thank you for your check. We have corrected Figure 4, as indicated.

Reviewer 4 Report

The authors report a new sensitive, economic, and specific method for monitoring of formaldehyde in food and cosmetics samples.  My first impression is excellent. Very well designed and performed an experiment with a wide area of application.

I do not have any critical remarks on the manuscript. It can be published the way it is.   

Author Response

“The authors report a new sensitive, economic, and specific method for monitoring of formaldehyde in food and cosmetics samples.  My first impression is excellent. Very well designed and performed an experiment with a wide area of application.

I do not have any critical remarks on the manuscript. It can be published the way it is.”

We are pleased that you have appreciate our paper. Thank you for your comments.

Round 2

Reviewer 2 Report

This manuscript developed a FA detection method. The sample preparation method is interesting. But the superiority was not presented and good results were not obtained. I think it is not suitable for the Journal.

Detail suggestions:

1: The results about the analytical method should be introduced in the abstract, such as LOD, LOQ, and the linear range.

2: In the conclusions, the linear range was listed as 0 – 40 ug/mL. It is very non-standard, and not rigorous. Some ultra-low concentration of FA can be detected? How about the LOD and LOQ?

3: Such as Fig. 7, these FA-spiked curves should not be listed in the text.

4: The proposed FA detection method should compare with other reported FA detection methods in a listed Table, which should be added in the text, and relative analysis method parameters and references should be presented.

5: The presentation of Figures should be improved.

 Author Response

Dear Reviewer,

thank you for your comments; you can find the answers to your comments  below:

1: The results about the analytical method should be introduced in the abstract, such as LOD, LOQ, and the linear range.

We have added that information in the abstract, as you suggest.

2: In the conclusions, the linear range was listed as 0 – 40 ug/mL. It is very non-standard, and not rigorous. Some ultra-low concentration of FA can be detected? How about the LOD and LOQ?

Thank you for your comment. The method we proposed was set up to obtain an exhaustive extraction of formaldehyde from various matrices with considerable content of said analyte. Therefore, the calibration curve was constructed based on the range of content reported in literature for the matrices investigated. As far as low concentrations are concerned, the method was not tested in those scenarios, and specific tests should be conducted: however, the advantages in recovery we observed would translate to those concentrations as well. We retain that, despite this investigation was not conducted, our work fits well in the Special Issue dedicated to innovative application of microextraction techniques, applied to complex matrices. We updated the text with the LOD and LOQ obtained in our conditions (highlighted in yellow in the main text).

3: Such as Fig. 7, these FA-spiked curves should not be listed in the text.

Thank you for your suggestion. We moved these two Figures in the Supplementary Materials, keeping the Table with the results in the main text.

4: The proposed FA detection method should compare with other reported FA detection methods in a listed Table, which should be added in the text, and relative analysis method parameters and references should be presented.

Thank you for your precious comment. We agree with you and in order to compare our work with similar ones, we have listed them and their main features in Table 4. Concerning the literatures used, we have only chosen studies that use on-sample derivatization of carbonyl compounds with PFBHA and HS extraction. Studies using direct injection, or other derivatizing reagents (such as DNPH), and procedures (such as on-fiber) were not taken into account in the comparison.

5: The presentation of Figures should be improved.

We have improved the figure presentation in the text, as much as possible, as you suggested.

In attachement, you'll find the main text with the integrations.